# CLICs-dependent chloride efflux is an essential and proximal upstream event for NLRP3 inflammasome activation

Tiantian Tang[1,2,3], Xueting Lang[1], Congfei Xu[1], Xiaqiong Wang[1], Tao Gong[1], Yanqing Yang[1], Jun Cui[4], Li Bai[1], Jun Wang[1], Wei Jiang[1,2] & Rongbin Zhou[1,2,3]

The NLRP3 inflammasome can sense different pathogens or danger signals, and has been reported to be involved in the development of many human diseases. Potassium efflux and mitochondrial damage are both reported to mediate NLRP3 inflammasome activation, but the underlying, orchestrating signaling events are still unclear. Here we show that chloride intracellular channels (CLIC) act downstream of the potassium efflux-mitochondrial reactive oxygen species (ROS) axis to promote NLRP3 inflammasome activation. NLRP3 agonists induce potassium efflux, which causes mitochondrial damage and ROS production. Mitochondrial ROS then induces the translocation of CLICs to the plasma membrane for the induction of chloride efflux to promote NEK7–NLRP3 interaction, inflammasome assembly, caspase-1 activation, and IL-1β secretion. Thus, our results identify CLICs-dependent chloride efflux as an essential and proximal upstream event for NLRP3 activation.

[1] Institute of Immunology and the CAS Key Laboratory of Innate Immunity and Chronic Disease, CAS center for Excellence in Molecular Cell Sciences, School of Life Sciences and Medical Center, University of Science and Technology of China, Hefei 230027, China. [2] Innovation Center for Cell Signalling Network, University of Science and Technology of China, Hefei 230027, China. [3] Hefei National Laboratory for Physical Sciences at Microscale, University of Science and Technology of China, Hefei 230027, China. [4] Key Laboratory of Gene Engineering of the Ministry of Education, State Key Laboratory of Biocontrol, School of Life Sciences, Sun Yat-sen University, Guangzhou 510080, China. Tiantian Tang and Xueting Lang contributed equally to this work. Wei Jiang and Rongbin Zhou jointly supervised this work. Correspondence and requests for materials should be addressed to W.J. (email: ustcjw@ustc.edu.cn) or to R.Z. (email: zrb1980@ustc.edu.cn)

The NLRP3 inflammasome is an intracellular protein complex formed by the innate immune sensor protein NLRP3, adapter protein ASC, and proteolysis enzyme caspase-1[1, 2]. This protein complex is assembled in response to microbial infection or various non-infectious stresses to promote the cleavage and activation of caspase-1, which induces the maturation and secretion of proinflammatory cytokines, such as IL-1β and IL-18[3–5]. The NLRP3 inflammasome has been reported to participate in the development of diverse inflammatory diseases, including type 2 diabetes, Alzheimer's disease, and gout[6–11], suggesting that NLRP3 might be a potential target for the treatment of inflammatory diseases. In addition, NLRP3 can be activated by various agonists with different characteristics or structures derived not only from pathogens, but also from environment stress, metabolic dysregulation, and tissue damage[1, 3]. Although mitochondrial dysfunction, reactive oxygen species (ROS) production, lysosome damage, or disturbance of intracellular ion homeostasis have been proposed for mediating NLRP3 activation by the above agonists[3, 12, 13], how these signaling events orchestrate NLRP3 inflammasome activation is still poorly understood.

The disturbance of intracellular ion homeostasis is a major cellular stress signal for NLRP3 inflammasome activation. Potassium efflux has been proposed to have an essential function in NLRP3 inflammasome activation because several NLRP3 activators reduce intracellular potassium concentration; by contrast, increasing the extracellular potassium concentration inhibits inflammasome activation by all tested NLRP3 activators[14, 15]. Although the cation channel P2RX7, which is a receptor for extracellular ATP[16], has a critical function in ATP-induced NLRP3 inflammasome activation or ATP-dependent non-canonical inflammasome activation[17, 18], the potassium channels responsible for other agonists-induced NLRP3 inflammasome activation have not been identified. Several studies suggest that calcium influx or mobilization is required for NLRP3 inflammasome activation, with TRPM2 and TRPV2 involved[19–22]. In addition, NLRP3 inflammasome activation can be blocked by several non-specific chloride channel inhibitors, and incubation of macrophages in chloride-free medium enhances ATP-induced caspase-1 activation and IL-1β production[22–25], suggesting that chloride current might contribute to NLRP3 inflammasome activation. However, how chloride current regulates NLRP3 inflammasome activation is unclear.

The chloride intracellular channel (CLIC) protein family consists of six evolutionary conserved proteins (CLIC1–CLIC6) and has been implicated in membrane remodeling, intracellular trafficking, vacuole formation, and actin reorganization[26, 27]. CLICs exist in both soluble and membrane-associated forms, and contain a putative transmembrane region and a nuclear localization signal, which are present in the N- and C-terminal domain, respectively[27]. CLICs are detected in both cytosol and intracellular organelles, including mitochondria, endosome, and nucleus[27–30]. CLICs often associate with the actin cytoskeleton and can undergo rapid redistribution between subcellular locations in dynamic actin-dependent trafficking events[27]. CLICs are structurally related to the omega-class of glutathione S-transferases and have intrinsic glutaredoxin-like enzymatic activity in vitro[31]. Under oxidative conditions, CLICs can undergo a reversible rearrangement of the GST-like fold and associate with artificial membranes and induce anion currents under non-reducing and low pH conditions[32–36], suggesting that soluble CLICs might translocate to the plasma membrane and form ion channels under specific conditions. Indeed, amyloid-β peptides can induce the translocation of CLIC1 to the plasma membrane and trigger CLIC1-dependent chloride current in microglia cells[37]. However, whether CLICs have ion channel activity under physiological conditions needs further validation. The functions of CLICs in innate immunity and inflammasome are largely unknown. Previous study has shown that CLIC4 has a function in innate immunity, because Clic4-deficient mice are resistant to LPS-induced septic shock[38], although the mechanisms are unclear.

In this study, we show that NLRP3 agonist-induced mitochondrial ROS promotes the membrane translocation of CLICs and chloride efflux, which then trigger NEK7–NLRP3 interaction and NLRP3 inflammasome activation. Thus, our results indicate that CLICs-dependent chloride efflux is an essential signaling event upstream of NLRP3 activation.

## Results

**IAA94 inhibits NLRP3 activation**. To assess the function of CLICs in NLRP3 inflammasome activation, we first examined whether indanyloxyacetic acid-94 (IAA94), which has shown inhibitory activity for CLICs[39], could inhibit NLRP3 inflammasome activation. When LPS-primed bone marrow-derived macrophages (BMDMs) were pretreated with IAA94 before nigericin challenge, caspase-1 activation and IL-1β maturation were suppressed by IAA94 in a dose-dependent manner (Fig. 1a, b), while the production of TNF, an inflammasome-independent cytokine, was not affected (Fig. 1c). To test whether this effect was specific to nigericin or common to NLRP3 agonists, we examined other agonists and found that IAA94 treatment also blocked ATP or monosodium urate crystals (MSU)-induced NLRP3 inflammasome activation (Fig. 1d, e). However, IAA94 had no effects on poly A:T transfection-induced AIM2 inflammasome activation or salmonella infection-induced NLRC4 inflammasome activation (Supplementary Fig. 1a–c). Moreover, nigericin-induced cell death was inhibited by IAA94 (Supplementary Fig. 1d), suggesting that IAA94 can block inflammasome-dependent pyroptosis. In addition, IAA94 inhibited NLRP3 inflammasome activation in human THP-1 cells (Fig. 1f). Taken together, these results demonstrate that IAA94 is a specific inhibitor for NLRP3 inflammasome.

**CLICs mediate NLRP3 inflammasome activation in macrophages**. To further confirm the effect of CLICs in NLRP3 inflammasome activation, we examined whether inhibition of CLICs expression suppressed NLRP3 activation. CLICs have six family members[26], only five members, including Clic1, 3, 4, 5, and 6, exist in mouse genome. We first checked the expression of the Clics in BMDMs and found that Clic1, Clic4, and Clic5 were expressed on BMDMs and LPS priming could upregulate the expression of Clic4 and Clic5, while had no effects on Clic1 expression (Supplementary Fig. 2a, b). In contrast, inflammasome stimulation had no effects on the expression of Clic1, Clic4, or Clic5 (Supplementary Fig. 2c). To determine which Clic protein was involved in NLRP3 activation, the expression of Clics was silenced by siRNA in BMDMs, respectively (Supplementary Fig. 2d, e). Knockdown each of these Clics had some inhibitory effects, but could not block nigericin-induced IL-1β production completely (Fig. 2a), suggesting the Clics have an important function in NLRP3 activation, but these proteins might have redundant functions.

To further confirm the function of Clics, we generated Clic1−/−, Clic4−/−, and Clic5−/− mice. The Clic4−/− mice were generated by homologous recombination (Supplementary Fig. 3a) and the Clic1−/− and Clic5−/− mice were generated using Crispr–Cas9 system. The deletion of proteins was confirmed in BMDMs by assessing the Clics expression (Supplementary Fig. 3b). Consistent with the siRNA-mediated knockdown, genetic deletion of each of these Clics only had moderate

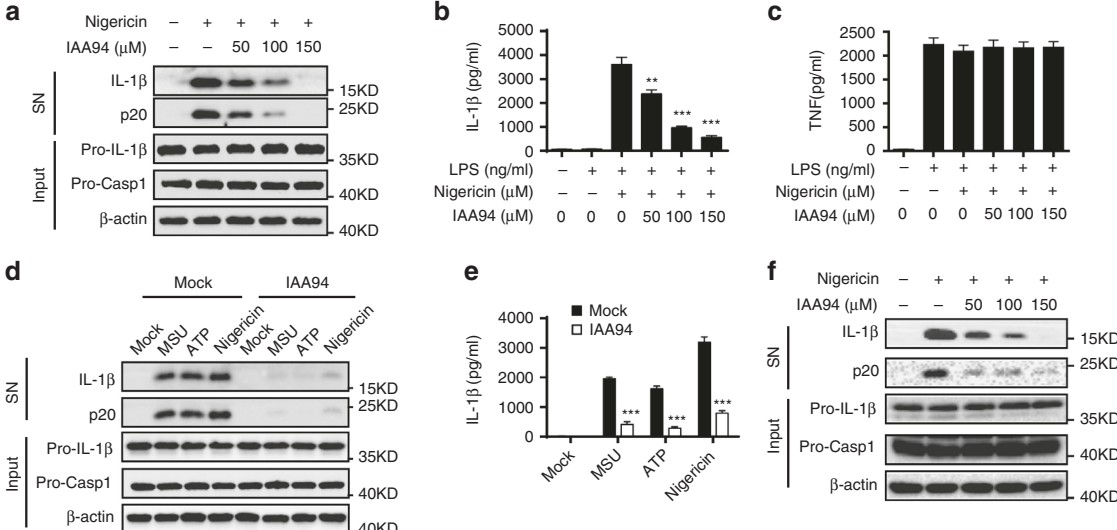

**Fig. 1** IAA94 inhibits NLRP3 inflammasome activation. **a** Immunoblot analysis of IL-1β and cleaved caspase-1 (p20) in culture supernatants (SN) of LPS-primed BMDMs treated with various doses (*above lanes*) of IAA94 and then left stimulated with nigericin, and immunoblot analysis of the precursors of IL-1β (pro-IL-1β) and caspase-1 (pro-caspase-1) in lysates of those cells (Input). **b**, **c** ELISA of IL-1β **b** and TNF **c** in supernatants from LPS-primed BMDMs treated with various doses of IAA94 and then stimulated with nigericin. **d**, **e** Immunoblot **d** or ELISA **e** analysis of IL-1β and cleaved caspase-1 (p20) in culture supernatants of LPS-primed BMDMs treated with of IAA94 (150 μM) and then left stimulated with MSU, nigericin, or ATP. **f** Immunoblot analysis of IL-1β and cleaved caspase-1 (p20) in supernatants from PMA-differentiated THP-1 cells treated with various doses (*above lanes*) of IAA94 and then left stimulated with nigericin. Data are from three independent experiments with biological duplicates in each (**b**, **c**, **e**; mean ± SEM of $n = 6$) or are representative of at least three independent experiments **a**, **d**, **f**. Two-way ANOVA **b** Student's *t*-test **e** **$^{**}P < 0.01$, $^{***}P < 0.001$

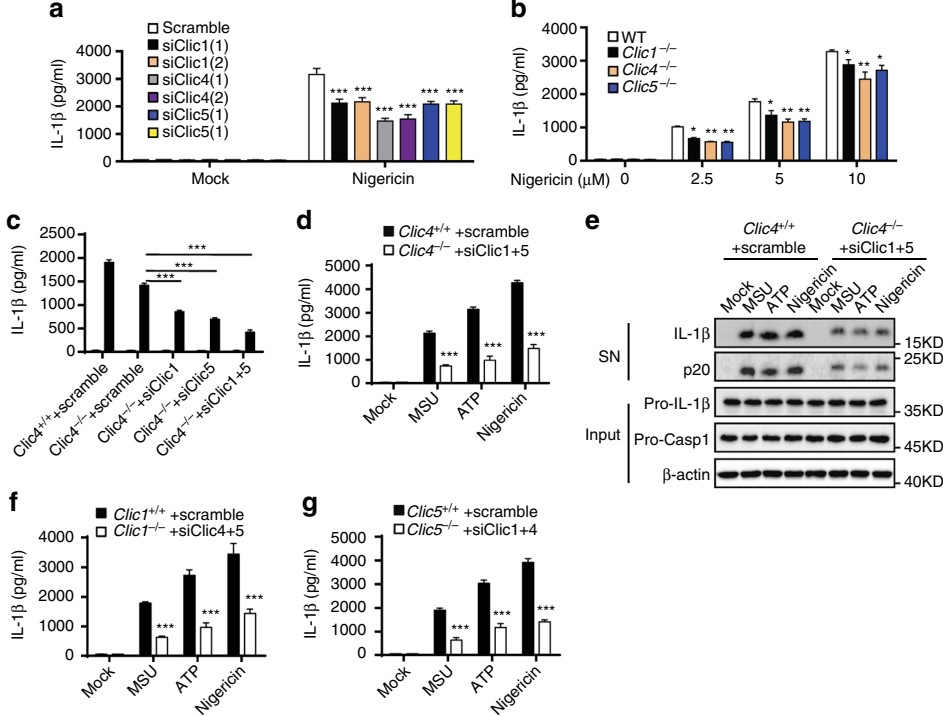

**Fig. 2** Inhibition of Clics suppresses NLRP3 inflammasome activation. **a** ELISA of IL-1β in culture supernatants of LPS-primed BMDMs transfected with siRNA against *Clic1, 4, or 5* and left stimulated with nigericin. **b** ELISA of IL-1β in culture supernatants of LPS-primed *Clic1$^{-/-}$*, *Clic4$^{-/-}$*, or *Clic5$^{-/-}$* BMDMs stimulated with nigericin. **c** ELISA analysis of IL-1β in culture supernatants of LPS-primed wildtype or *Clic4$^{-/-}$* BMDMs transfected with siRNA against *Clic1, Clic5*, or both of them and left stimulated with nigericin. **d**, **e** ELISA **d** or Immunoblot **e** analysis of IL-1β in culture supernatants of LPS-primed wildtype or *Clic4$^{-/-}$* BMDMs transfected with siRNA against *Clic1* and *Clic5*, and left stimulated with nigericin, MSU, or ATP. **f** ELISA of IL-1β in culture supernatants of LPS-primed wildtype or *Clic1$^{-/-}$* BMDMs transfected with siRNA against *Clic4* and *Clic5*, and left stimulated with nigericin, MSU, or ATP. **g** ELISA of IL-1β in culture supernatants of LPS-primed wild type or *Clic5$^{-/-}$* BMDMs transfected with siRNA against *Clic1* and *Clic4* and left stimulated with nigericin, MSU, or ATP. Data are from three independent experiments with biological duplicates in each (**a**–**d**, **f**, **g**; mean ± SEM of $n = 6$) or are representative of three independent experiments **e**. Student's *t*-test **a**–**d**, **f**, **g**, **$^{**}P < 0.01$, $^{***}P < 0.001$

inhibitory effects on low dose of nigericin-induced IL-1β production (Fig. 2b). The effects of *Clics* genetic deletion on IL-1β production were even worse than siRNA-mediated knockdown when BMDMs were treated with high doses of nigericin (Fig. 2b), suggesting that genetic deletion of a family member of *Clics* might cause compensatory upregulation of other family members. Indeed, we observed the compensatory upregulation of other *Clic* family members in BMDMs when single-*Clic* gene was deleted (Supplementary Fig. 3c–e). To overcome the redundancy of Clic members, we tried to obtain triple knockout mice, but deletion of two members of the three genes caused embryonic lethality. We then tried to decrease the expression of *Clic1*, *Clic5*, or both of them in *Clic4*$^{-/-}$ BMDMs using siRNA. Although the efficiency of siRNA was compromised because of the compensatory upregulation of *Clic1* and *Clic5*, siRNA transfection successfully repressed their expression and did not cause cell death in *Clic4*$^{-/-}$ BMDMs (Supplementary Fig. 3c, f). We found that inhibition of *Clic1* or *Clic5* expression could decrease nigericin-induced IL-1β production in *Clic4*$^{-/-}$ BMDMs (Fig. 2c). Inhibition of both *Clic1* and *Clic5* expression in *Clic4*$^{-/-}$ BMDMs then further decreased nigericin-induced IL-1β production (Fig. 2c). Indeed, inhibition of both *Clic1* and *Clic5* expression in *Clic4*$^{-/-}$ BMDMs could significantly suppressed MSU, nigericin, or ATP-induced caspase-1 activation and IL-1β production (Fig. 2d, e). The similar results were obtained in *Clic1*$^{-/-}$ or *Clic5*$^{-/-}$ BMDMs when treated with siRNA against the other two *Clic* genes (Fig. 2f, g and Supplementary Fig. 3d, e). These results suggest that Clics have redundant functions in NLRP3 inflammasome activation. In addition, our results showed that inhibition of *Clic1* and *Clic5* in *Clic4*$^{-/-}$ BMDMs could significantly inhibited cytosolic LPS-induced non-canonical NLRP3 inflammasome activation, but had no effects on cytosolic dsDNA-induced AIM2 inflammasome or salmonella infection-induced NLRC4 inflammasome activation (Supplementary Fig. 4a–c). Our results also showed that Clics were not required for LPS-induced priming signaling, and had no effects on LPS-induced NLRP3, pro-IL-1β expression, and inflammasome-independent cytokines production (Supplementary Fig. 4d–f).

Thus, these results indicate that CLICs have an important function for NLRP3 inflammasome activation in macrophages.

**CLICs mediate NLRP3 inflammasome activation in vivo.** Since CLICs mediate NLRP3 inflammasome activation in macrophages, we then analyzed the function of Clics in NLRP3-dependent inflammation in vivo. Intraperitoneal injection of MSU elicited an NLRP3-dependent inflammatory response in the mouse peritoneum characterized by IL-1β production and massive neutrophil influx[11]. Consistent with the inhibitory function of IAA94 for NLRP3 inflammasome activation in macrophages, IAA94 treatment efficiently suppressed MSU injection-induced IL-1β production and neutrophil influx in vivo (Fig. 3a, b and Supplementary Fig. 5). To overcome the compensatory effects of Clic5 in *Clic4*$^{-/-}$ mice, we delivered siRNA against *Clic5* to the peritoneal cavity of mice by using nanoparticle to inhibit its expression (Fig. 3c). Compared with wild-type mice, *Clic4*$^{-/-}$ mice had less neutrophil influx and IL-1β production after injection of MSU, which were further compromised by siClic5 delivery in *Clic4*$^{-/-}$ mice (Fig. 3d, e). These results support that CLICs are important for NLRP3 inflammasome activation.

**CLICs mediate NLRP3 activation via promoting chloride efflux.** We then investigated how Clics regulated NLRP3 inflammasome activation. We first tested whether Clics could directly interact with the components of NLRP3 inflammasome, including NLRP3, ASC, or NEK7[40–42]. The results showed that nigericin or ATP treatment could not induce the recruitment of Clics to NEK7–NLRP3 or NLRP3–ASC complex (Supplementary Fig. 6a, b), suggesting that Clics were not present in NEK7–NLRP3–ASC complex during inflammasome activation.

Since CLICs function as chloride channels to regulate ion homeostasis in cells and blockade of the chloride efflux by chemical inhibitors can suppress NLRP3 inflammasome activation[23, 26]. CLICs may mediate NLRP3 inflammasome activation by promoting chloride efflux. First, we found that

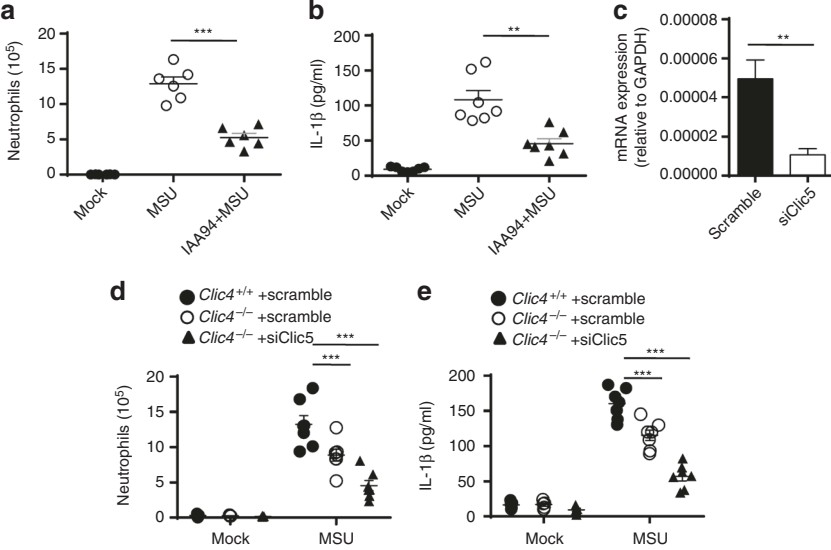

**Fig. 3** Clics are important for NLRP3 inflammasome activation in vivo. **a**, **b** FACS analysis of neutrophil numbers **a** or ELISA **b** of IL-1β in the peritoneal cavity of C57BL/6J mice intraperitoneally injected with MSU (1 mg per mouse) with or without IAA94 (50 mg/kg of body weight). n = 6 or 7 per group. **c** qPCR analysis of *Clic5* expression in peritoneal macrophages from *Clic4*$^{-/-}$ mice intraperitoneally injected with nanoparticle-encapsulated siRNA (40 μg per mouse). **d**, **e** FACS analysis of neutrophil numbers **d** or ELISA **e** of IL-1β in the peritoneal cavity of *Clic4*$^{-/-}$ mice intraperitoneally injected with nanoparticle-encapsulated siRNA (40 μg per mouse) and MSU (1 mg per mouse). n = 5–7 per group. Data are shown as mean ± SEM and are representative of two independent experiments. Student's *t*-test, **P < 0.01, ***P < 0.001

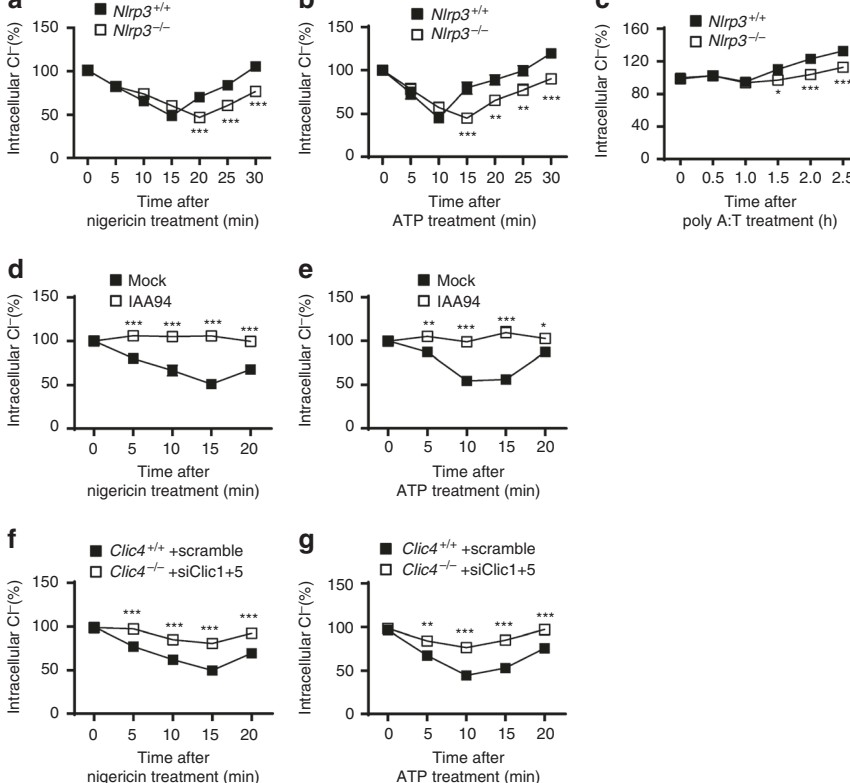

**Fig. 4** Clics-dependent chloride efflux during NLRP3 inflammasome activation. **a–c** Qualification of the decrease of intracellular chloride in wild type or $Nlrp3^{-/-}$ BMDMs at different time points after nigericin **a**, ATP **b**, or poly A:T **c** treatment. **d**, **e** Qualification of the decrease of intracellular chloride in BMDMs at different time points after nigericin **d** or ATP **e** treatment with or without the pretreatment with IAA94 (150 µM). **f**, **g** Qualification of the decrease of intracellular chloride in $Clic4^{-/-}$ BMDMs transfected siRNA against $Clic1$ and $Clic5$ at different time points after nigericin **f** or ATP **g** treatment. Data are from three independent experiments with biological duplicates in each (mean ± SEM of $n = 6$). Student's $t$-test, $*P < 0.05$, $**P < 0.01$, $***P < 0.001$

NLRP3 agonists, such as ATP and nigericin, induced the chloride efflux in BMDMs rapidly (Fig. 4a, b), suggesting NLRP3 agonists trigger chloride efflux. In addition, NLRP3 agonists could induce chloride efflux in $Nlrp3^{-/-}$ cells, although it happened a little slower compared with wild-type cells, indicating that chloride efflux is an upstream signaling event of NLRP3 inflammasome activation (Fig. 4a, b). We also found that the intracellular chloride in BMDMs recovered at later time points after treatment (Fig. 4a, b), this might explain why chloride efflux was not observed when macrophages were treated with nigericin for 30 min in a previous study[14]. This recovery might be caused by the loss of membrane potential or membrane integrity induced by inflammasome activators which resulted in the high concentration of extracellular chloride to reenter cells. In contrast, AIM2 agonist could not induce significant chloride efflux (Fig. 4c). Importantly, NLRP3 agonists-induced chloride efflux could be inhibited by IAA94 treatment (Fig. 4d, e). We also found that anthracene-9-carboxylic acid (A9C), another inhibitor for CLICs[43], could also inhibit NLRP3 agonists-induced chloride efflux and IL-1β production (Supplementary Fig. 7a, b). In contrast, chloride channel blocker 4,4′-diisothiocyano-2,2′ stilbene-disulfonic acid (DIDS), which has no CLICs inhibitory activity[43], could not suppress NLRP3 agonists-induced chloride efflux (Supplementary Fig. 7c). However, DIDS could inhibit nigericin-induced IL-1β production, suggesting that it might target the downstream event of CLICs-dependent chloride efflux to inhibit NLRP3 activation (Supplementary Fig. 7d). Moreover, inhibition of Clics expression could also inhibit NLRP3 agonists-induced chloride efflux in macrophages (Fig. 4f, g). These results

indicate that NLRP3 agonists-induced chloride efflux in macrophages depends on CLICs.

To further investigate the function of CLICs-dependent chloride efflux in NLRP3 activation, we then examined whether replacement of chloride with other anions in culture medium to facilitate chloride efflux could promote NLRP3 inflammasome activation. Previous study has shown that extracellular chloride can limit the ability of ATP to gate the P2RX7 and inhibit ATP-induced IL-1β production, so substitution of extracellular chloride with gluconate can enhance ATP-induced IL-1β production[24]. We then investigated whether this effect was specific to ATP or was common to all NLRP3 agonists. Indeed, we found that substitution of extracellular chloride with gluconate also enhanced MSU or nigericin-induced IL-1β production, but had no effects on AIM2 or NLRC4 inflammasome activation (Fig. 5a–c and Supplementary Fig. 8a, b). Since MSU or nigericin-induced IL-1β production is P2RX7-independent[17], these results suggest the extracellular chloride not only regulates P2RX7-dependent inflammasome activation, but also functions as a common regulator for NLRP3 inflammasome. To further confirm this, LPS-primed BMDMs were incubated in buffers in which chloride was replaced by different anions, respectively, including iodide, bromide, glutamate, and gluconate. The results showed that incubation of BMDMs in chloride-free buffers for longer time resulted in chloride efflux and spontaneous IL-1β production without the presence of NLRP3 agonists (Fig. 5d and Supplementary Fig. 8c). Moreover, chloride-free buffers-induced caspase-1 activation and IL-1β production were blocked by $Nlrp3$ deletion (Fig. 5e, f), indicating that chloride efflux might be

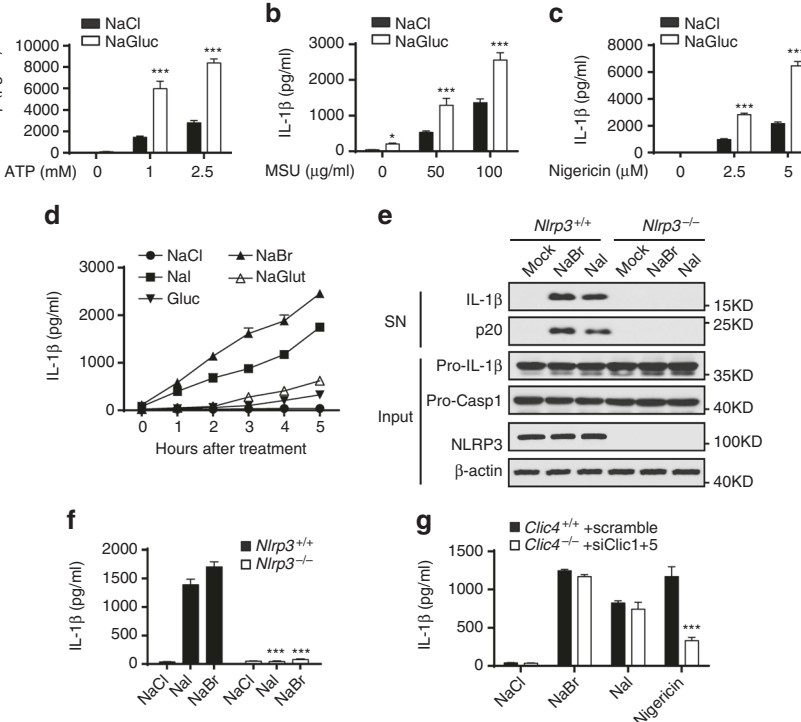

**Fig. 5** Chloride efflux is essential for NLRP3 inflammasome activation. **a–c** ELISA of IL-1β in LPS-primed BMDMs which were transferred to a basic NaCl saline (130 mM NaCl, 5 mM KCl, 1 mM MgCl$_2$, 1 mM CaCl$_2$, 20 mM HEPES (pH 7.4), 1 mg/ml BSA, 10 mM glucose) or chloride-free NaGluc saline (130 mM NaGluc, 5 mM KGluc, 1 mM MgGluc$_2$, 1 mM CaGluc$_2$, 20 mM HEPES (pH 7.4), 1 mg/ml BSA, 10 mM glucose) and left stimulated with ATP **a**, MSU **b**, or nigericin **c**. **d** ELISA of IL-1β in LPS-primed BMDMs which were transferred to NaCl saline, NaBr saline (130 mM NaBr, 5 mM KBr, 1 mM MgBr$_2$, 1 mM CaBr$_2$, 20 mM HEPES (pH 7.4), 1 mg/ml BSA, 10 mM glucose), NaI saline (130 mM NaI, 5 mM KI, 1 mM MgI$_2$, 1 mM CaI$_2$, 20 mM HEPES (pH 7.4), 1 mg/ml BSA, 10 mM glucose), NaGluc saline or NaGlut (130 mM NaGlut, 5 mM KGlut, 1 mM MgGlut$_2$, 1 mM CaGlut$_2$, 20 mM HEPES (pH 7.4), 1 mg/ml BSA, 10 mM glucose) saline for different time points. **e, f** Immunoblot analysis IL-1β and cleaved caspase-1 **e** or ELISA of IL-1β in supernatants **f** of LPS-primed Nlrp3$^{+/+}$ or Nlrp3$^{-/-}$ BMDMs which transferred to NaCl, NaBr or NaI saline for 3 h. **g** ELISA of IL-1β in supernatants of LPS-primed Clic4$^{-/-}$ BMDMs transfected siRNA against Clic1 and Clic5 and left stimulated with nigericin for 30 min or transferred to NaCl, NaBr or NaI saline for 3 h. Data are from three independent experiments with biological duplicates in each (**a–d**, **f**, **g**; mean ± SEM of n = 6) or are representative of three independent experiments **e**. Student's t-test **a–c**, **f**, **g**, ***P < 0.001

sufficient to activate NLRP3 inflammasome. It should be noted that, in contrast with nigericin, chloride-free buffers-induced IL-1β production was independent of Clics (Fig. 5g), suggesting that other chloride channels are involved in this condition. Taken together, these results demonstrate the essential function of CLICs in NLRP3 inflammasome activation by promoting chloride efflux.

**CLICs act downstream of K efflux and mitochondrial ROS.** To further clarify how CLICs-mediated chloride efflux regulates NLRP3 activation, we then studied whether CLICs affected potassium efflux, which was an upstream signaling event of NLRP3 activation[14, 15]. First, inhibition of Clics expression or activity had no impact on ATP or nigericin-induced potassium efflux (Fig. 6a, b and Supplementary Fig. 9a). Second, potassium-free buffer treatment caused the decrease of intracelluar chloride before IL-1β production (Supplementary Fig. 9b, c). Third, potassium-free buffer-induced IL-1β production was suppressed by inhibition of Clics expression or activity (Fig. 6c and Supplementary Fig. 9d). In addition, we found that IAA94 had no effects on ATP-induced calcium influx (Supplementary Fig. 9e). These results indicate that CLICs and chloride efflux act downstream of potassium efflux to activate NLRP3 inflammasome.

Mitochondria damage, represented as mitochondria fission, clustering, and ROS production, is another upstream signaling event for NLRP3 activation[12]. Nigericin-induced mitochondrial

damage and ROS production were not affected in BMDMs when the activity or expression of Clics were inhibited (Fig. 6d and Supplementary Fig. 10a, b). Incubation of BMDMs in chloride-free buffers could not induce obvious mitochondria damage or ROS production (Supplementary Fig. 10c). Moreover, treatment with Manganese (III) tetrakis (4-benzoic acid)porphyrin chloride (MnTBAP), which mimics mitochondrial superoxide dismutase, inhibited nigericin, or potassium-free buffer-induced IL-1β production, but almost had no effects on chloride-free buffer-induced IL-1β production (Fig. 6e). These results indicate that CLICs-dependent chloride efflux is a downstream event of mitochondrial damage and ROS production during NLRP3 inflammasome activation.

Although both potassium efflux and mitochondrial dysregulation are involved in NLRP3 inflammasome activation, the relationship between these two events is not clear. Consistent with a previous report[14], inhibition of mitochondrial ROS had no effects on nigericin-induced potassium efflux (Supplementary Fig. 11a), suggesting that ROS production is not at the upstream of potassium efflux. However, incubation of BMDMs in potassium-free buffer resulted in strong mitochondrial damage and ROS production (Supplementary Fig. 11b). Consistent with this, IL-1β secretion in potassium-free buffer was inhibited by mitochondrial ROS inhibitor (Supplementary Fig. 11c). These results indicate that potassium efflux induces mitochondrial damage and ROS production to promote NLRP3 inflammasome activation.

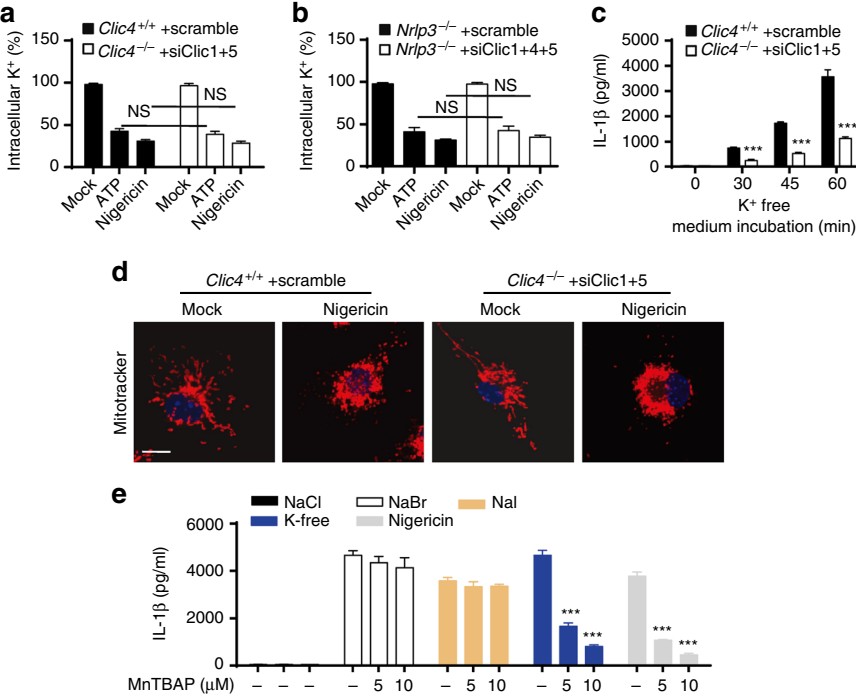

**Fig. 6** Clics act downstream of potassium efflux or mitochondrial damage to promote NLRP3 activation. **a**, **b** Qualification of the decrease of intracellular potassium in LPS-primed $Clic4^{-/-}$ BMDMs transfected with siRNA against Clic1 and Clic5 **a** or $Nlrp3^{-/-}$ BMDMs tranfected with siRNA against Clic1, Clic4 and Clic5 **b** and left stimulated with ATP or nigericin. **c** ELISA of IL-1β in supernatants of LPS-primed $Clic4^{-/-}$ BMDMs transfected siRNA against Clic1 and Clic5 and left transferred to potassium-free saline (135 mM NaCl, 1 mM MgCl$_2$, 1 mM CaCl$_2$, 20 mM HEPES (pH 7.4), 1 mg/ml BSA, 10 mM glucose) for 3 h. **d** Confocal microscopy analysis of LPS-primed $Clic4^{-/-}$ BMDMs transfected with siRNA against Clic1 and Clic5 and then left stimulated with nigericin, followed by staining with Mitotracker red and DAPI. **e** ELISA of IL-1β in supernatants of LPS-primed BMDMs transfected stimulated with nigericin or transferred to NaBr, NaI saline or potassium-free saline with or without the presence of indicated doses of MnTBAP. Data are from three independent experiments with biological duplicates in each (**a–c**, **e**; mean ± SEM of $n = 6$) or are representative of three independent experiments **d**. Two-way ANOVA **e**, Student's $t$-test **a–c**, ***$P < 0.001$, $NS$ not significant

**NLRP3 agonists induce plasma membrane translocation of CLICs**. Since mitochondrial damage and ROS production are the proximity events of CLICs-dependent chloride efflux during NLRP3 inflammasome activation, we then investigated whether ROS-promoted CLICs-dependent chloride efflux. Indeed, oxidative stress can promote CLICs to undergo rearrangement of the protein structure and then translocate to cell membrane to promote chloride efflux[32–35, 37, 44]. Indeed, we found that nigericin treatment could induce a rapid enrichment of Clic1, Clic4, and Clic5 in plasma membrane fraction of BMDMs (Fig. 7a). Moreover, NLRP3 agonists-induced enrichment of Clic1, Clic4, and Clic5 in plasma membrane could be suppressed by MnTBAP (Fig. 7b, c). We further tried to confirm the plasma membrane translocation of Clic1, 4, and 5 by microscopy analysis, but the antibodies against Clic1 or Clic5 were not working for immunofluorescence. The ROS-dependent translocation of Clic4 to plasma membrane was confirmed by microscopy analysis of the distribution of Clic4 in BMDMs treated with ATP or nigericin, but not with poly A:T (Fig. 7d–g and Supplementary Fig. 12). Consistent with these results, MnTBAP also suppressed ATP or nigericin-induced chloride efflux (Fig. 7h, i). Thus, these results suggest that ROS can promote the plasma membrane translocation of CLICs to promote chloride efflux and NLRP3 activation.

**CLICs are Important for NEK7–NLRP3 interaction**. CLICs-dependent chloride efflux acts downstream of potassium efflux and mitochondrial damage to activate NLRP3 inflammasome, suggesting that chloride efflux is an upstream event proximal to NLRP3 inflammasome assembly. We then investigated the exact function of CLICs-dependent chloride efflux in inflammasome assembly. First, nigericin-induced ASC oligomerization was impaired when Clics expression were inhibited (Fig. 8a), indicating that chloride efflux affects ASC activation or NLRP3–ASC interaction. Moreover, nigericin-induced NLRP3–ASC interaction was also suppressed by inhibition of Clics (Fig. 8b). Recently, NEK7 has been proposed as an essential component of NLRP3 inflammasome and can regulate NLRP3 oligomerization and inflammasome assembly[40–42]. Indeed, the interaction between NEK7 and NLRP3 during inflammasome activation was also impaired by inhibition of Clics expression (Fig. 8c). These results suggest that CLICs-dependent chloride efflux regulates NEK7–NLRP3 complex formation to promote NLRP3 inflammasome activation.

**Discussion**

The NLRP3 inflammasome plays a critical function in innate immunity, inflammation, and several chronic inflammation associated diseases, but the mechanisms underlying its activation are complicated[1, 2]. The disturbance of intracellular ion homeostasis is a key upstream event for NLRP3 activation[14, 15, 23, 24], but the ion channels mediating ions flow need to be identified and the mechanisms of ions regulate NLRP3 inflammasome activation are not clear. Here we demonstrate that CLICs-dependent chloride efflux acts downstream of potassium efflux and mitochondrial ROS production to activate NLRP3 inflammasome, suggesting that chloride efflux is a proximal upstream event for NLRP3 inflammasome assembly and activation.

Potassium efflux, mitochondrial ROS, and chloride efflux are upstream events of NLRP3 inflammasome activation[12, 14, 15, 23, 24],

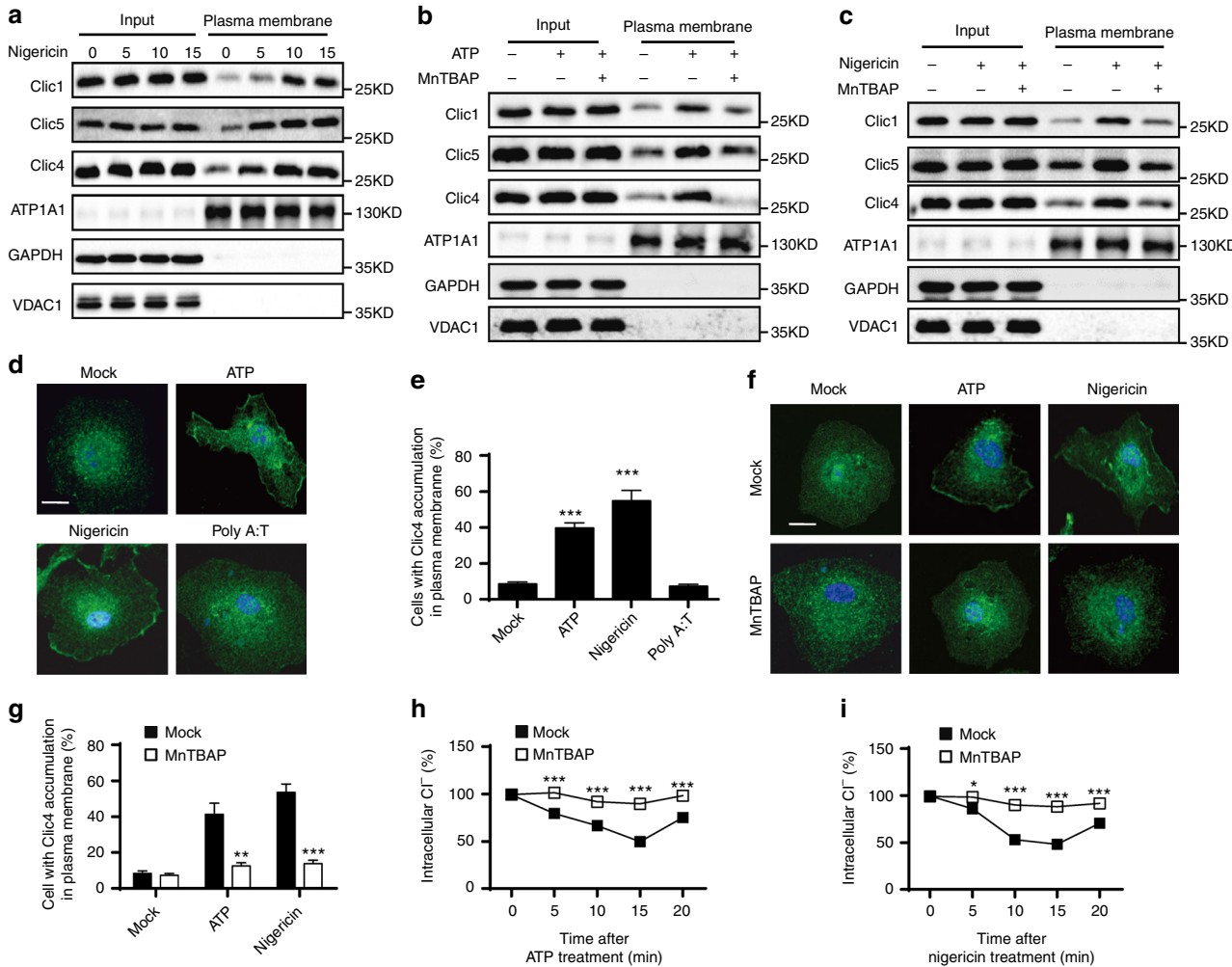

**Fig. 7** ROS promotes NLRP3 agonists-induced Clics translocation and chloride efflux. **a** Immunoblot analysis of the indicated proteins in total lysates (Input) or isolated plasma membrane of LPS-primed BMDMs stimulated with nigericin at different time points. **b**, **c** Immunoblot analysis of the indicated proteins in total lysates (Input) or isolated plasma membrane of LPS-primed BMDMs stimulated with ATP **b** or nigericin **c** for 15 min with or without the presence of MnTBAP (10 μM). **d**, **e** Confocal microscopy analysis **d** or qualification of cells with membrane translocation of Clic4 **e** in BMDMs stimulated with ATP, nigericin or poly A:T for 15 min. *Scale bar*, 10 μm. **f**, **g** Confocal microscopy analysis **f** or qualification of cells with membrane translocation of Clic4 **g** in BMDMs stimulated with ATP or nigericin for 15 min with or without the presence of MnTBAP (10 μM). *Scale bar*, 10 μm. **h**, **i** Qualification of the decrease of intracellular chloride of BMDMs stimulated with ATP **h** or nigericin **i** with or without the presence of MnTBAP (10 μM). Data are from three independent experiments with biological duplicates in each (**h**, **i**; mean ± SEM of $n = 6$) or are representative of three independent experiments **a–d**, **f**. Student's *t*-test **g–i**, *$P < 0.05$, **$P < 0.01$, ***$P < 0.001$

but how these events orchestrate NLRP3 inflammasome assembly is not understood. Consistent with previous report[14], ROS scavengers could not block nigericin-induced potassium efflux. Inversely, free potassium buffer could induce mitochondrial ROS production and ROS scavengers could block free potassium buffer-induced IL-1β production. These results suggest that potassium efflux is an upstream event of mitochondria damage and ROS production during NLRP3 activation. Our further studies showed that CLICs acted downstream of ROS, which promoted the membrane translocation of CLICs and triggered chloride efflux. Thus these results suggest potassium efflux-mitochondrial damage-CLICs-chloride efflux as the signaling chain upstream of NLRP3 inflammasome activation (Fig. 8d)

Our study demonstrates NLRP3 agonists induce the translocation of CLICs to plasma membrane and robust chloride efflux via CLICs-dependent manner, suggesting that CLICs themselves might function as chloride channels to mediate NLRP3 inflammasome activation via promoting chloride efflux. Consistent with this, CLIC members have chloride channel activity when they

exist as membrane form[34, 35, 37, 44]. Although previous studies have shown that the channels formed by recombinant CLICs in artificial bilayers have poorly selectivity and are almost equally permeable by potassium and chloride[45, 46], suppression of the expression or activity of CLICs had no effects on potassium efflux or calcium influx during NLRP3 inflammasome activation, suggesting that CLICs might function as specific chloride channels under this condition. However, it should be noted that we cannot exclude the possibility that CLICs might function as an activator or modulator for a membrane chloride channel during inflammasome activation. Recently, the volume-regulated anion channel (VRAC) has been proposed to be involved in NLRP3 inflammasome activation because its chemical inhibitor can block NLRP3 inflammasome activation[23], suggesting CLICs might function as an activator for VRAC. So, if the function of VRAC in NLRP3 inflammasome activation could be confirmed by genetic evidences, the relationship between VRAC and CLICs needs to be clarified in future. In addition, although we have observed the translocation of CLICs to plasma membrane, CLICs-dependent

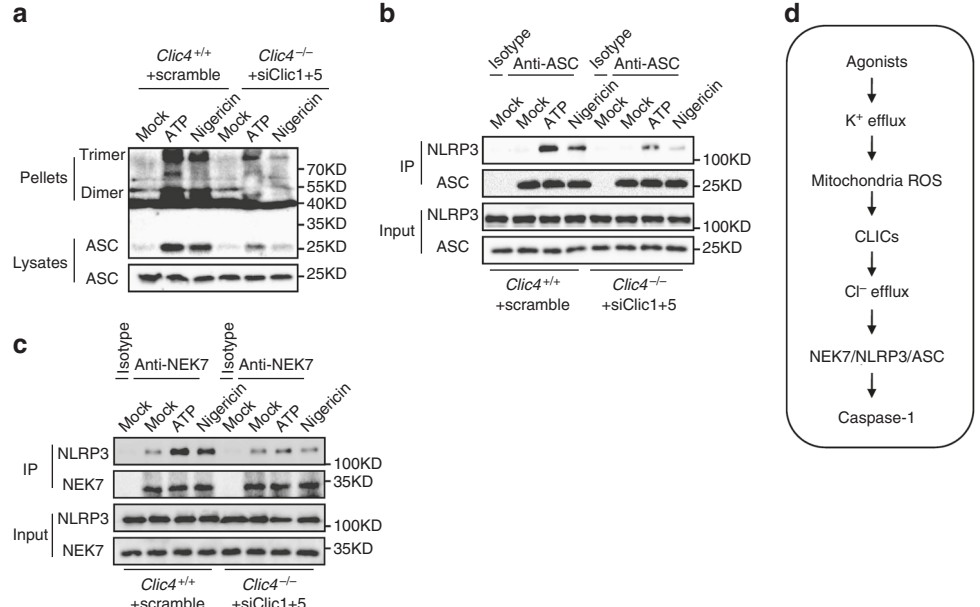

**Fig. 8** Clics are important for NLRP3 inflammasome assembly. **a** Immunoblot analysis of ASC oligomerization in lysates of LPS-primed *Clic4$^{-/-}$* BMDMs transfected with siRNA against *Clic1* and *Clic5* and left stimulated with ATP or nigericin. **b** Immunoprecipitation (IP) and immunoblot analysis of the interaction of endogenous NLRP3 and ASC in LPS-primed *Clic4$^{-/-}$* BMDMs transfected with siRNA against *Clic1* and *Clic5*, and left stimulated with ATP or nigericin. **c** IP and immunoblot analysis of the interaction of endogenous NLRP3 and NEK7 in LPS-primed *Clic4$^{-/-}$* BMDMs transfected with siRNA against *Clic1* and *Clic5*, and left stimulated with ATP or nigericin. **d** Model for the function and mechanism of Clics and chloride efflux in NLRP3 inflammasome activation. Data are representative of three independent experiments **a–c**

chloride efflux and the inhibitory effects of IAA94 on IL-1β production during NLRP3 inflammasome activation, the cytosolic CLICs might also contribute to NLRP3 inflammasome activation, because IAA94 can also inhibit the glutaredoxin-like enzymatic activity of soluble CLICs[31]. We then tested whether soluble CLICs were present in NLRP3 inflammasome complex during activation, but we did not find the interactions between CLICs with the known NLRP3 inflammasome components, suggesting that CLICs or their glutaredoxin-like enzymatic activity cannot directly regulate the assembly of NLRP3 inflammasome. However, we still could not exclude the possible indirect contribution of cytosolic CLICs in NLRP3 inflammasome activation.

Although our results have shown that IAA94 can block NLRP3 inflammasome activation, suppression of the expression of CLICs cannot inhibit NLRP3 inflammasome activation completely. One possibility is that the family members are functionally redundant. Indeed, inhibition of all *Clic1, 4*, and *5* expression had much better inhibitory effects for inflammasome activation than inhibition of single-*Clics* expression, possibly because the expression of other Clics were compensatorily increased. The redundant functions of CLIC family members have also been described in previous studies[47, 48]. Another evidence for the functional redundancy of Clics was that deletion of two members of *Clic1, Clic4*, or *Clic5* caused embryonic lethality, but deletion of one member did not, suggesting that CLICs are redundant in embryonic development. Another possibility is that other IAA94 sensitive chloride channels are also involved in NLRP3 inflammasome activation.

Our study demonstrates that CLICs have a specific function to regulate NLRP3 inflammasome activation. Inhibition of Clics expression had no effects on AIM2 or NLRC4 inflammasome activation. In addition, inhibition of Clics expression also had no effects on inflammasome priming, such as LPS-induced NLRP3 or pro-IL-1β expression. A previous study has shown that LPS-induced inflammasome-independent cytokines production, such as TNF or IL-6, were slightly impaired in *Clic4$^{-/-}$* macrophages[38],

although the mechanisms for this were not clear, because they also showed that LPS-induced NF-κB or MAPK activation were normal[38]. However, another study showed the normal TNF production in LPS/IFN-γ treated *Clic4$^{-/-}$* macrophages[49]. Consistent with this result, our study showed that inhibition of Clics expression could not compromise LPS-induced TNF or IL-6 production.

Our study demonstrates that CLICs-dependent chloride efflux can regulate NLRP3 inflammasome assembly. Inhibition of Clics activity or expression suppressed NLPR3–NEK7 interaction and the subsequent NLRP3–ASC complex formation and ASC oligomerization, suggesting that CLICs and intracellular chloride might regulate NEK7–NLRP3 interaction to affect NLRP3 inflammasome activation. As intracellular chloride homeostasis is critical for many cell functions including cell-signaling transduction[50, 51], intracellular chloride might function as a signaling messenger to regulate NEK7–NLRP3 interaction directly or indirectly. The detailed mechanisms of how intracellular chloride regulates inflammasome assembly need to be further investigated.

Thus, our results reveal that CLICs-dependent chloride efflux acts downstream of potassium efflux-mitochondrial ROS axis and functions as a proximal upstream event for NLRP3 inflammasome assembly and suggest the possibility to target CLICs to treat NLRP3-driven diseases.

## Methods

**Mice.** For the generation of *Clic4$^{-/-}$* mice, a targeting vector was designed to replace exon 2 of the *Nlrp3* gene with a neomycin resistance (neo). The construct was electroporated into C57BL/6-derived embryonic stem (ES) cells. Correctly targeted ES cells were injected into blastocysts and the resulting chimeric animals were crossed to C57BL/6 mice to obtain the heterozygous F1 mice. The heterozygous F1 mice were crossed to generate homozygous and wild-type littermates. *Clic1$^{-/-}$* and *Clic5$^{-/-}$* mice were generated using CRISPR–Cas9 approaches as described[52]. In brief, the vectors encoding Cas9 (44758, Addgene) and guide RNA were in vitro transcribed into messenger RNA and gRNA followed by injection into the fertilized eggs that were transplanted into pseudo-pregnant mice. The targeted

genome of F0 mice was amplified with PCR and sequenced and the chimeras were crossed with wild-type C57BL/6 mice to obtain the heterozygous F1 mice. The F1 mice were further crossed with wild-type C57BL/6 mice for at least three generations. Mice were genotyped by PCR analysis followed by sequencing and the resulted heterozygous mice were crossed to generate homozygous and wild-type littermates. All mice are in C57BL/6 background and are harbored in the specific pathogen-free facility in University of Science and Technology of China. Sex-matched littermates of the mutant mice were used as control. At the end of experiments, all mice were killed by $CO_2$ inhalation. All animal experiments were approved by the Ethics Committee of University of Science and Technology of China.

**Reagents**. Phorbol-12-myristate-13-acetate (PMA), MSU, ATP, nigericin, poly A: T, Pam3CSK4, A9C, DIDS, and IAA94 were purchased from Sigma. MitoSox, MitoTracker Red, DAPI, MQAE, Lipofectamine RNAiMAX, Lipofectamine 2000, and M-MLV were from Invitrogen. Ultrapure LPS was obtained from Invivogen. SYRB Green was from TransGen Biotech. MnTBAP was from Santa Cruz. *Salmonella* is a gift from R.V. Bruggen. Anti-human cleaved IL-1β (2021, 1:1000 dilution) and anti-human caspase-1(2225, 1:1000 dilution) antibodies were purchased from Cell Signalling. Anti-human pro-IL-1β (60136-1-lg, 1:1000 dilution) anti-CLIC1(14545-1-AP, 1:500 dilution), anti-GAPDH (10494-1-AP, 1:1000 dilution), anti-VDAC1 (66345-1-lg, 1:1000 dilution), and anti-ATP1A1 (55187-1-AP, 1:1000 dilution) antibodies were from Proteintech. Anti-CLIC4 (SC-135739, 1:1000 dilution), anti-NEK7 (SC-50756, 1:500 dilution), and anti-ASC (rabbit, SC-22514, 1:400 dilution) antibodies were from Santa Cruz. Anti-ASC (mouse, 04–147, 1:1000 dilution) antibody was from Merck Millipore. Anti-CLIC5 (ACL-025, 1:250 dilution) antibody was from Alomone labs. Anti-mouse IL-1β (AF-401-NA, 1:1000 dilution) antibody was from R&D. Anti-mouse caspase-1 (p20) (AG-20B-0042, 1:1000 dilution) and anti-NLRP3 (AG-20B-0014, 1:1000 dilution) antibodies were from Adipogen. Anti-β-actin (P30002, 1:1000 dilution) antibody was from Abmart.

**Cell preparation and stimulation**. Human THP-1 from American Type Culture Collection cells were grown in RPMI 1640 medium, supplemented with 10% FBS and 50 μM 2-mercaptoethanol. THP-1 cells were routinely tested for mycoplasma contamination. THP-1 cell were differentiated for 3 h with 100 nM PMA and then incubated overnight. Bone-marrow macrophages were isolated and cultured as described[53]. To induce NLRP3 inflammasome activation, $5 \times 10^5 \text{ml}^{-1}$ macrophages were plated in 12-well plates. The following morning, the medium was replaced and cells were stimulated with 50 ng ml$^{-1}$ LPS or 400 ng ml$^{-1}$ Pam3CSK4 (for non-canonical inflammasome activation) for 3 h. After that, the cells were treated with pharmacologic inhibitors for 30 min, and then stimulated for 4 h with MSU (150 μg ml$^{-1}$), S. typhimurium (multiplicity of infection (MOI)) or for 30 min with ATP (2.5 mM) or nigericin (10 μM). Cells were transfected with poly A:T (0.5 μg ml$^{-1}$) for 4 h or LPS (500 ng ml$^{-1}$) overnight by using Lipofectamine 2000 (Invitrogen). Cell extracts and precipitated supernatants were analyzed by immunoblot.

**Immunofluorescence**. BMDMs were plated on coverslips overnight and then used for stimulation or staining with Mitotracker red (50 nM) or MitoSox (5 μM). After washing three times with PBST, the cells were fixed with PFA 4% in PBS for 15 min and then washed three times with PBST. After permeabilization with Triton X-100 and blocking with 10% goat serum in PBS, cells were incubated with anti-Clic4 antibody (1:200 dilution in 10% goat serum) overnight at 4 °C. After washing with PBST, cells were incubated with secondary antibodies (Invitrogen) in 10% goat serum-PBS for 60 min and rinsed in PBST. Confocal microscopy analyses were carried out using a Zeiss LSM700. For the qualification of Clic4 translocation to membrane, scanning filed were randomly picked, and at least 100 cells were counted in each slides.

**siRNA-mediated gene silences in BMDMs**. BMDMs were plated in 12-well plates ($4 \times 10^5$ cells per well) and then were transfected with 100 nM siRNA using Lipofectamine RNAiMAX (Invitrogen) according to the manufacturer's guidelines. See Supplementary Table 1 for siRNA sequences.

**ELISA**. Supernatants from cell culture or peritoneal lavage fluid were assayed for mouse TNF, IL-6, and IL-1β (R&D) according to manufacturer's instructions.

**Assay for cytosolic calcium**. BMDMs were primed with LPS (50 ng ml$^{-1}$) and then treated with IAA94 (150 μM) for 30 min. After that, the cells were incubated with fluo-3-AM (2 mM) for 15 min, and then each well was briefly washed and resuspended with ice-cold PBS. After that, the cells were stimulated with ATP (2.5 mM), and then assessed by flow cytometry (BD) for analysis of the concentration of calcium.

**Real time PCR**. BMDMs were dissolved in Trizol reagent (Invitrogen). cDNA was synthesized from extracted total-RNA using M-MLV Reverse Transcriptase kit (Invitrogen, 28025021) according to the manufacturer's protocol. Quantitative PCR was performed with SYBR-Green premix (TransGen Biotech) and detected by a Real Time PCR System (StepOne, Applied Biosystems). *GAPDH* was used as an internal control gene.

**MSU-induced peritonitis**. MSU-induced peritonitis was induced by intraperitoneal injection of 1 mg MSU crystals dissolved in 0.5 ml sterile PBS. After 6 h, mice were killed by exposure to $CO_2$ and peritoneal cavities were washed with 10 ml cold PBS. Peritoneal lavage fluid was assessed by flow cytometry (BD) with the neutrophil markers Gr-1 and CD11b for analysis of the recruitment of polymorphonuclear neutrophils and determined IL-1β production by ELISA.

**Treatment of mice with nanoparticle-encapsulated siRNA**. To specifically silence the expression of *Clic5* in peritoneal macrophages, a cationic lipid–assisted ePEG-PLA nanoparticle was used to encapsulate siRNA[54]. The nanoparticle-encapsulated siRNA (40 μg per mouse) was injected intraperitoneally into mice. After 3 days, these mice were used for detecting Clic5 expression in peritoneal macrophages or further experiments.

**Determination of intracellular potassium and chloride**. Intracellular potassium measurements were performed by inductively coupled plasma optical emission spectrometry with a PerkinElmer Optima 7300 DV spectrometer. The culture media of 6-well plates were removed, cells were washed in potassium-free buffer (139 mM NaCl, 1.7 mM NaH$_2$PO$_4$, and 10 mM Na$_2$HPO$_4$, pH 7.2), and 10 ml of ultrapure HNO$_3$ were added. Samples were transferred to glass bottles and boiled for 30 min at 100 °C, Add ddH$_2$O to 5 ml. For accurate measurement of the intracellular chloride, the supernatants of 12-well plates were removed, add ddH$_2$O (200 μl per well) and kept 15 min at 37 °C. The lysates were transferred to 1.5 ml EP tube, and centrifuged at 10,000×g for 5 min. 50 μl supernatants were then transferred to a new 1.5 ml EP tube and mixed with 50 μl MQAE (10 μM). The absorbance was tested by BioTek Multi-Mode Microplate Readers (Synergy2). For both of the measurements, a control was settled in every experiment to determine the extracellular amount of chloride remaining after aspiration, and this value was subtracted.

**Isolation of plasma membrane fraction**. BMDMs ($2 \times 10^7$) were plated in 10-cm-treated cell culture dish and the medium were changed to opti-MEM in the following morning, and then the cells were primed with ultra-LPS (50 ng ml$^{-1}$) for 9 h. After that, the cells were stimulated with 2 mM ATP or 2 μM nigericin at different time points. The supernatants were removed, and cells were washed three times using ice-cold PBS. The plasma membrane were extracted and purified for Plasma Membrane Protein Extraction Kit (Abcam, ab65400) according to manufacturer's instructions.

**Immunoprecipitation**. BMDMs were primed with LPS (50 ng ml$^{-1}$) and then stimulated with ATP or nigericin for 30 min. After that, the cells were resuspended in lysis buffer (50 mM Tris, pH 7.8, 50 mM NaCl, 1% (vol/vol) Nonidet-P40, 5 mM EDTA, and 10% (vol/vol) glycerol) and the cell lyates were immunoprecipitated with anti-ASC (0.5 μg ml$^{-1}$), anti-NEK7 (1 μg ml$^{-1}$) antibody, or control IgG at 4 °C and protein A/G agarose resin for overnight. The samples were washed in lysis buffer for five times and then were used for immunoblot analysis.

**ASC oligomerization assay**. BMDMs were washed in ice-cold PBS, and 500 μl of ice-cold NP-40 (50 mM Tris, pH 7.8, 50 mM NaCl, 1% (vol/vol) Nonidet-P40, 5 mM EDTA, and 10% (vol/vol) glycerol) was added. Cells were dissolved on the shaker for 30 min at 4 °C and then centrifuged at 330×g for 10 min at 4 °C. The pellets were washed twice in 1 ml of ice-cold PBS and resuspended in 500 μl of ice-cold PBS. 2 mM disuccinimydylsuberate (Sangon biotech, C100015) was added to the resuspended pellets, which were incubated for 30 min with rotation at room temperature. Samples were then centrifuged at 330×g for 10 min at 4 °C. The supernatants were removed, and the cross-linked pellets were resuspended in 30 μl of sample buffer. Samples were boiled for 10 min at 101 °C and analyzed by western blotting.

**Statistical analyses**. All values are expressed as the mean ± SEM. Statistical analysis were performed with the unpaired $t$-test for two groups or two-way ANOVA (GraphPad Software) using for multiple groups with all data points showing a normal distribution. The genotype of the mice or cells was not blinded. No exclusion of data points was used. Sample sizes were selected on the basis of preliminary results to ensure an adequate power. $P < 0.05$ were considered significant.

**Date availability**. The data that support this study are available within the article and its Supplementary Information files or available from the authors upon request.

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

## Acknowledgements

This work was supported by National Basic Research Program of China (2014CB910800), NSFC (81330078, 81525013, 81571609), the Young Talent Support Program, the Strategic Priority Research Program of the Chinese Academy of Sciences

(XDPB03), the China Postdoctoral Science Foundation, the Young Talent fund of China Association for Science and Technology and the Fundamental Research Funds for the Central Universities.

## Author contributions

T.T., T.L., C.X., X.W., and T.G. performed the experiments of this work; T.T., L.B., J.W., J.C., W.J., and R.Z. designed the research. T.T., T.L., W.J., and R.Z. wrote the manuscript. W.J. and R.Z. supervised the project.

## Additional information

**Competing interests:** The authors declare no competing financial interests.

