## [Peer Review File · Nature Communications]

Reviewers' comments:

Reviewer #1 (Remarks to the Author):

The paper by Tang and coworkers reports on the tentative identification of a novel mechanism coupling potassium efflux to NLRP3 inflammasome activation. It is now an established fact that potassium efflux is a (the) main intracellular signal triggering inflammasome activation and IL-1 processing, however how such an ion change promotes inflammasome assembly is as yet a mystery. Therefore, the study by Tang et al is both important and timely. Nevertheless, I have some reservations that decrease the overall appeal of this study.

Major criticisms

1. First of all, some statements (assumptions) reported in the paper are incorrect and may indicate a rather superficial knowledge of the relevant literature. For example, Introduction, second para, "....., but the responsible potassium channels have not been identified". This is not true since a major pathway for the potassium efflux responsible for NLRP3 inflammasome activation has been identified with the P2X7 receptor (that incidentally is at times misspelt in the text, i.e. "...R2X7R"). The P2X7-KO mouse is unable to activate the NLRP3 inflammasome in response to most stimuli, and we now know that also the atypical caspase-11 inflammasome depends on P2X7 for its activity (see recent papers by Gabriel Nunez and co-workers).

2. IAA94 is used at a concentration (50 to 150 μ M) previously reported to inhibit intracellular chloride channels. This is however a rather high concentration. It is necessary to check whether these IAA94 doses have any effect on cell viability, and therefore measurement of a few basic indicators of cell damage is suggested.

3. Although to my knowledge there is no report on the effects of IAA-94 on pannexin-1, it is well known that several chloride channel blockers inhibit this plasma membrane channel. Can the Authors exclude that IAA-94 is not blocking pannexin-1 activity? This is relevant in view of the general assumption that pannexin-1 is one of the plasma membrane pathways mediating the intracellular ion changes that eventually trigger NLRP3 inflammasome activation. Incidentally, I also strongly suggest to check the effect of IAA-94 on the P2X7 receptor: several channel blockers also block this receptor/channel at high concentrations.

4. The protocol for stimulation of IL-1 processing and release is crucial for the analysis of inflammasome activation, however I don't see anywhere a detailed description. For example, what was the ATP (or MSU or nigericin) concentration used for the experiments described in Figure 1? For how long was the ATP (or nigericin or MSU) stimulation carried out?

5. In several blots (e.g. panels A and D) the β -actin band is not shown. Is there any reason for this?

6. It is intriguing that neither CLIC silencing or genetic deletion had a substantial effect on NLRP3 inflammasome activation. A blot showing CLIC protein levels in silenced cells is missing. In addition, it is not clear to me what "CLICs expression" refers to in Figure S2E: is it the sum of the expression levels of mRNAs for all CLICs? This is confusing. Incidentally, I don't understand why the Authors state that ".....inhibition of Clic1 and Clic5 in Clic-/- BMDMs inhibited cytosolic LPS-induced non-canonical NLRP3 inflammasome activation". Where are the data supporting this statement?

7. I am intrigued by the data shown in Panels A and B of Figure 4. Nigericin and ATP-triggered chloride efflux is very similar, if not identical. This is surprising to me because the mechanism whereby nigericin and ATP cause potassium efflux is completely different: the former is a potassium ionophore, while the latter opens a cation-selective channel followed by the opening of a large conductance non-selective pore. There is something that I do not understand in these results. I understand even less the blocking effect of IAA-94 on the ATP-dependent chloride efflux: opening of the ATP-activated large conductance non-selective pore allows transmembrane fluxes of organic ions as large as lucifer yellow, and of ATP itself, thus I do not understand why chloride efflux should be blocked!!

8. It is not clear what the Authors mean by stating that ".....these results suggest that chloride efflux not only regulates the binding of ATP to R2X7R (incidentally, it should be P2X7R)" . As I understand the data shown in this MS, chloride efflux is supposed to be downhill to P2X7R activation, not uphill.
9. Contrary to the Authors' statement that CLIC inhibition or genetic deletion protect mitochondria from nigericin-dependent damage, images shown in Figure 6D and S6 clearly show mitochondria are clumped and swollen.
10. Last but not least: usage of the English language is poor, there are many misspelt words and awkward constructios.

Reviewer #2 (Remarks to the Author):

Comments to Authors

This report describes the involvement of CLIC proteins in the assembly of NLRP3 and subsequent activation of the inflammasome as measured by release of IL-1B. The authors use a combination of drug inhibitors and activators combined with gene knockout and siRNA strategies to make a logical story that the assembly and activation of the inflammasome requires intracellular ionic modification including reduced potassium leading to mitochondrial mediated ROS and reduction of intracellular chloride possibly mediated by 3 CLIC proteins (1,4,5) in macrophages (BMDC). While this message is supported by the data presented, additional data would strength the conclusions.

Comments and suggestions:

1. The introduction is brief and contains the most important details about NLRP3 inflammasome activation and the factors that were found to contribute to such a process. However, there was not enough information about the activity of the soluble form of the CLIC proteins and their distribution in cells. This protein family is not widely known and there was no mention about the intracellular distribution, other functions and factors that contribute to the ion channel formation of the CLIC proteins. Oxoreductive enzymatic functions have recently been shown for CLIC proteins (Al Khamici et al, PLOS ONE, 2015) and this function could be especially relevant for the current work. Having some of this information will give the readers more context to evaluate the results and the final conclusions.

2. Fig 1: The authors state that IAA94 is an inhibitor for the CLIC proteins. This statement is too specific for the CLICs as IAA94 can be an inhibitor for other chloride ion channels. Also it would be important to study the other chloride ion channel blockers that were tested on the function of CLIC proteins, such as A9C and DIDs. As previous studies have shown that IAA94 and A9C were able to block the ion channel activity as well as the enzymatic function of CLIC1 protein, but not DIDS, this trio of inhibitors has discrimination value. Additional concerns center on the viability of cells that were treated with up to 150uM of IAA94 and also the viability of cells in which 3 CLIC proteins are knocked out or down. Some test for viability would be helpful.

3. Fig 4: Graphs A, B and even C show an intracellular chloride ion percentage more than 100% after 30 mins or 2.5 hours post-Nigericin, ATP and A:T treatment. Is this a rebounds effect or within the variation of the measurement? This is a bit confusing since the graphs do not show error bars and if

we assume the error bars are small, then why do we see a small significant difference between the Mock and IAA94 20 mins post ATP treatment (graph E) but there is no significant difference between NLRP3 and its mutant after 2.5 hours treatment with poly A:T (graph C)?

4. Fig 7: It would be useful to show the change in input under the same treatments as shown for the plasma membranes to judge overall changes for each protein. This set of experiments would also be enhanced if duplicated by CLIC protein immunofluorescence also, since it might detect where the translocated CLIC proteins travel from. Previously CLIC 1, 4 and 5 have been described in various cellular compartments including mitochondria so that the additional information gained from IIF could be very informative. A time course study is also essential to understand the dynamics of CLIC4 migrating to the plasma membrane and the rapid change in chloride seen over a matter of minutes.

5. Fig 8: Here again additional very informative information could be obtained if the authors showed that a CLIC protein was actually present in the NLRP3-NEK7 complex using their IP methods. Such a discovery would add a new dimension to our understanding of this protein family.

Comments on the Discussion section:

For the sake of full disclosure, it would be important to reveal that, according to previous work, CLIC1 is a poorly anion selective ion channel and CLIC4 and CLIC5 are non selective ion channels with equal permeability to potassium and chloride. Therefore, it is important to include the possibility of other IAA94 sensitive chloride channels having involvement in this study.

To reiterate an alternative explanation for the results, an analysis of the time course of membrane translocation is very important. Depending on these data, the majority of the results in this paper may have been obtained from the non-membrane fractions of cells (e.g. IAA94 that was included in Fig1, was also able to block the enzymatic function of the soluble form of CLIC1 and presumably of the other CLIC proteins. Even the immunofluorescence imaging data provided does not show the CLICs in the membrane fractions of cells. Again it is critical to talk about the possibility of linking the enzymatic function of the CLIC proteins to NLRP3 activation and assembly process.

Finally, there are numerous typographical errors in the manuscript that will require careful editorial corrections.

Reviewer #3 (Remarks to the Author):

Tang et al. address the role of chloride intracellular channels (CLIC's) in Nlrp3 inflammasome activation and report that CLIC's play a crucial role downstream of K⁺ and ROS signaling but upstream to NEK7-Nlrp3 complex formation and caspase-1 activation. They also demonstrate that subsequent to mitochondrial ROS generation, CLIC's translocate to the plasma membrane promoting chloride efflux, which results in NEK7-Nlrp3-ASC association. The study is well performed and conclusion appropriately drawn. I do however have a few concerns:

1) LPS induced NFκB and IRF3 were shown to upregulate the expression levels of clic4 (abundantly expressed on macrophages) in major organs employing a positive feed forward loop (Ref 28) but the data presented in figure S2A contradicts or shows minimal effects on the expression levels. How do authors explain this and would it be possible for authors to present protein levels of clic1/4/5 under LPS-priming and NLRP3 inflammasome activation conditions.

- 2) Do soluble CLIC's present in the cytosol interact with NEK7-ASC-Nlrp3 complex?
- 3) The specificity of IAA94 is a concern as it potentially blocks K⁺ and Ca²⁺ signaling in addition to CLICs. This should be discussed.
- 4) Data from figures 2 and 3 suggest that CLIC4 & 5 play a dominant role. Have the authors looked at Clc4^{-/-} macrophages in which either Clc1 or Clc5 have been singly targeted with siRNA. Although CLIC1/4 double knockouts are embryonically lethal, is this also the case for CLIC4/5 or CLIC1/5? I realize the generation of new double knockout mice may be beyond the scope of the current study the authors should at least try to address this with an in vitro siRNA approach.
- 5) Why didn't the authors use siRNA against CLIC1 in their in vivo model (figure3)? How do authors justify their claim of in vivo data with regard to compensatory effects of CLIC1 overexpression (figure S2E) in the absence of CLIC4?
- 6) In figure 5G the chloride-free buffers induced IL-1b independent of CLICs leading to the hypothesis that other chloride channels are involved. This should be substantiated by evaluating intracellular Cl⁻ levels to demonstrate a Cl⁻ efflux under these conditions. In addition, incubation of BMDMs (Fig S6C) in Cl⁻ free buffers did not induce any mitochondrial ROS production; do the authors propose that Cl⁻ efflux is not only required but also sufficient for NLRP3 inflammasome activation.
- 7) The authors should provide additional details in the methods section regarding the specifics for timing of addition of pharmacologic inhibitors and for the co-immunoprecipitation technique so these studies can be replicated by others.

REVIEWERS' COMMENTS:

Reviewer #1 (Remarks to the Author):

I am satisfied with the rebuttal.

Reviewer #2 (Remarks to the Author):

no further comments

Reviewer #3 (Remarks to the Author):

The authors have addressed my previous concerns. I have only one comment: the authors might consider including the data showing IAA94 treatment doesn't affect Ca²⁺ signaling (reviewer 3 Point#3) in supplement information as this is particularly relevant for the current study as previously calcium was shown to regulate CLIC4 expression in mouse and human keratinocytes.

Reviewer #1 (Remarks to the Author):

The paper by Tang and coworkers reports on the tentative identification of a novel mechanism coupling potassium efflux to NLRP3 inflammasome activation. It is now an established fact that potassium efflux is a (the) main intracellular signal triggering inflammasome activation and IL-1 processing, however how such an ion change promotes inflammasome assembly is as yet a mystery. Therefore, the study by Tang et al is both important and timely. Nevertheless, I have some reservations that decrease the overall appeal of this study.

Major criticisms

1. First of all, some statements (assumptions) reported in the paper are incorrect and may indicate a rather superficial knowledge of the relevant literature. For example, Introduction, second para, "... , but the responsible potassium channels have not been identified". This is not true since a major pathway for the potassium efflux responsible for NLRP3 inflammasome activation has been identified with the P2X7 receptor (that incidentally is at times misspelt in the text, i.e. "...R2X7R"). The P2X7-KO mouse is unable to activate the NLRP3 inflammasome in response to most stimuli, and we now know that also the atypical caspase-11 inflammasome depends on P2X7 for its activity (see recent papers by Gabriel Nunez and co-workers).

Reply: Thanks for the comments.

1) In current literature, P2RX7 is only responsible for ATP-induced NLRP3 inflammasome activation or ATP-dependent noncanonical inflammasome activation, but is not involved in MSU, nigericin, silica-induced NLRP3 inflammasome activation (Gustin et al, 2015, Plos One; Karmakar et al, 2016, Nat Commu; Iyer et al, 2009, PNAS). So we modified the text as "Although the cation channel P2RX7, which is a receptor for extracellular ATP¹⁶, plays a critical role in ATP-induced NLRP3 inflammasome activation or ATP-dependent noncanonical inflammasome activation^{17, 18}, the potassium channels responsible for other agonists-induced NLRP3 inflammasome activation have not been identified" in the Introduction.

2) In addition, the "P2X7R" was corrected as "P2RX7" the in revised manuscript.

2. IAA94 is used at a concentration (50 to 150 uM) previously reported to inhibit intracellular chloride channels. This is however a rather high concentration. It is necessary to check whether these IAA94 doses have any effect on cell viability, and therefore measurement of a few basic indicators of cell damage is suggested.

Reply: Thanks very much for the suggestion. We examined the effects of IAA94 on cell viability and found that IAA94 itself could not cause significant cell death at these concentrations, but it could inhibit nigericin-induced pyroptosis (supplementary Fig.1d in the revised manuscript).

3. Although to my knowledge there is no report on the effects of IAA-94 on pannexin-1, it is well known that several chloride channel blockers inhibit this plasma membrane channel. Can the Authors exclude that IAA-94 is not blocking pannexin-1 activity? This is relevant in view of the general assumption that pannexin-1 is one of the plasma membrane pathways mediating the intracellular ion changes that eventually trigger NLRP3 inflammasome activation. Incidentally, I also strongly suggest to check the effect of IAA-94 on the P2X7 receptor: several channel blockers also block this receptor/channel at high concentrations.

Reply: Thanks very much for the comments.

1) Although the early report suggest that pannexin-1 might be involved in ATP-induced NLRP3 inflammasome activation, but the later reports using pannexin-1 KO cells found that pannexin-1 is not involved in ATP-induced pore formation and inflammasome activation (Yan et al, 2011, J Immunol; Wang et al, 2013, Protein Cell). We also confirmed these results using BMDMs from pannexin-1 KO mice and found that deletion of pannexin-1 had not effects on ATP-induced IL-1b production, potassium efflux, chloride efflux and the membrane permeability to Yo-PRO-1 (see the figure below). These results suggest that exclude the possibility that IAA94 inhibit NLRP3 activation via blocking the activity of pannexin-1.

2) In addition, we also found that IAA94 could not inhibit ATP-induced potassium efflux (supplementary Fig.8a), suggesting that IAA94 had no effect on the activity of P2RX7.

4. The protocol for stimulation of IL-1 processing and release is crucial for the analysis of inflammasome activation, however I don't see anywhere a detailed description. For example, what was the ATP (or MSU or nigericin) concentration used for the experiments described in Figure 1? For how long was the ATP (or nigericin or MSU) stimulation carried out?

Reply: Thanks very much for the suggestion. The detailed protocol has been added in the Method and Materials in the revised manuscript.

5. In several blots (e.g. panels A and D) the β-actin band is not shown. Is there any reason for this?

Reply: Thanks very much for the comments. According to our experiences, the pro-caspase-1 expression is stable during inflammasome activation, so we used it as a control in the original submission. In the revised manuscript, we added the β-actin in the figures by providing new data or re-blotting the samples.

6. It is intriguing that neither CLIC silencing or genetic deletion had a substantial effect on NLRP3 inflammasome activation. A blot showing CLIC protein levels in silenced cells is missing. In addition, it is not clear to me what "CLICs expression" refers to in Figure S2E: is it the sum of the expression levels of mRNAs for all CLICs? This is confusing. Incidentally, I don't understand

why the Authors state that “.....inhibition of Clic1 and Clic5 in Clic^{-/-} BMDMs inhibited cytosolic LPS-induced non-canonical NLRP3 inflammasome activation”. Where are the data supporting this statement?

Reply: Thanks very much for the suggestions. 1) The blot showing CLIC protein levels in silenced cells were provided in supplementary Fig.2e. 2) "The CLICs expression" refers to in Figure 2E means the expression of each CLIC family member, not the sum of the expression levels of mRNAs for all CLICs. We apologized for the confusing and the text has been modified in the revised manuscript. 3) The conclusion " inhibition of Clic1 and Clic5 in Clic^{-/-} BMDMs inhibited cytosolic LPS-induced non-canonical NLRP3 inflammasome activation" was supported by the data shown in supplementary Fig.4a (the Fig.S3a in the original manuscript).

7. I am intrigued by the data shown in Panels A and B of Figure 4. Nigericin and ATP-triggered chloride efflux is very similar, if not identical. This is surprising to me because the mechanism whereby nigericin and ATP cause potassium efflux is completely different: the former is a potassium ionophore, while the latter opens a cation-selective channel followed by the opening of a large conductance non-selective pore. There is something that I do not understand in these results. I understand even less the blocking effect of IAA-94 on the ATP-dependent chloride efflux: opening of the ATP-activated large conductance non-selective pore allows transmembrane fluxes of organic ions as large as lucifer yellow, and of ATP itself, thus I do not understand why chloride efflux should be blocked!!

Reply: Thanks for the comments.

1) Although the mechanisms of ATP or nigericin-induced potassium efflux are different, both of them induced potassium efflux were rapid (Muñoz-Planillo et al, 2013, Immunity). In addition, both ATP and nigericin could induced similar CLICs translocation to membrane (Fig.7). Similarly, potassium free buffer also induced a similar chloride efflux (supplementary Fig.8c). These could explain why nigericin and ATP-triggered chloride efflux is very similar.

2) We examined whether IAA94 could affect the formation of the non-selective pore and found that IAA94 treatment or suppression of the expression of CLICs could inhibit the intake of YO-PRO-1 (see the figure below), but could not inhibit the potassium efflux (Fig.6a, b and supplementary Fig.8a). These results suggest that CLICs might contribute to ATP-induced non-selective pore formation, although the mechanism need to be clarified in future.

8. It is not clear what the Authors mean by stating that “.....these results suggest that chloride efflux not only regulates the binding of ATP to R2X7R (incidentally, it should be P2X7R)”. As I understand the data shown in this MS, chloride efflux is supposed to be downhill to P2X7R activation, not uphill.

Reply: We apologized for the confusing. Our results found that P2RX7 activation promoted chloride efflux (Fig.5b). The sentence has been modified as "Since MSU or nigericin-induced IL-1 β production is P2RX7-independent¹⁷, these results suggest the extracellular chloride not only regulates P2RX7-dependent inflammasome activation, but also functions as a common regulator for NLRP3 inflammasome ".

9. Contrary to the Authors' statement that CLIC inhibition or genetic deletion protect mitochondria from nigericin-dependent damage, images shown in Figure 6D and S6 clearly show mitochondria are clumped and swollen.

Reply: Thanks for the comments. We didn't claim that CLIC inhibition or genetic deletion protect mitochondria from nigericin-dependent damage. In the original manuscript, we stated that "Nigericin-induced mitochondrial damage and ROS production were normal in BMDMs when the activity or expression of Clics were inhibited (Fig. 6D and Fig. S6A, B) ". It seems this sentence is confusing, so we modified this sentence as "Nigericin-induced mitochondrial damage and ROS production were not affected in BMDMs when the activity or expression of Clics were inhibited (Fig.6d and supplementary Fig.9a, b)" in the revised manuscript.

10. Last but not least: usage of the English language is poor, there are many misspelt words and awkward constructios.

Reply: Thanks for the suggestion. We have carefully checked and modified the English language.

Reviewer #2 (Remarks to the Author):

Comments to Authors

This report describes the involvement of CLIC proteins in the assembly of NLRP3 and subsequent activation of the inflammasome as measured by release of IL-1B. The authors use a combination of drug inhibitors and activators combined with gene knockout and siRNA strategies to make a logical story that the assembly and activation of the inflammasome requires intracellular ionic modification including reduced potassium leading to mitochondrial mediated ROS and reduction of intracellular chloride possibly mediated by 3 CLIC proteins (1,4,5) in macrophages (BMDC). While this message is supported by the data presented, additional data would strength the conclusions.

Comments and suggestions:

1. The introduction is brief and contains the most important details about NLRP3 inflammasome activation and the factors that were found to contribute to such a process. However, there was not enough information about the activity of the soluble form of the CLIC proteins and their distribution in cells. This protein family is not widely known and there was no mention about the intracellular distribution, other functions and factors that contribute to the ion channel formation of the CLIC proteins. Oxoreductive enzymatic functions have recently been shown for CLIC proteins (Al Khamici et al, PLOS ONE, 2015) and this function could be especially relevant for

the current work. Having some of this information will give the readers more context to evaluate the results and the final conclusions.

Reply: Thanks for the comments and suggestions. We have added these information about CLICs in the Introduction of the revised manuscript: "The chloride intracellular channel (CLIC) protein family consists of six evolutionary conserved proteins (CLIC1–CLIC6) and has been implicated in membrane remodeling, intracellular trafficking, vacuole formation, actin reorganization and other processes^{26, 27}. CLICs exist in both soluble and membrane-associated forms and contain a putative transmembrane region (PTM) and a nuclear localization signal (NLS), which are present in the N- and C-terminal domain respectively²⁷. CLICs have been detected in both cytosol and intracellular organelles, including mitochondria, endosome and nuclear²⁷⁻³⁰. CLICs often associate with the actin cytoskeleton and can undergo rapid redistribution between subcellular locations in dynamic actin-dependent trafficking events²⁷. CLICs are structurally related to the omega-class of glutathione S-transferases (GSTOs) and have intrinsic glutaredoxin-like enzymatic activity *in vitro*³¹. Under oxidative conditions, CLICs can undergo a reversible rearrangement of the GST-like fold and associate with artificial membranes and induce anion currents under nonreducing and low pH conditions³²⁻³⁶, suggesting that soluble CLICs might translocate to plasma membrane and form ion channels under certain conditions. Indeed, amyloid- β (A β) peptides can induce the translocation of CLIC1 to plasma membrane and trigger CLIC1-dependent chloride current in microglia cells³⁷. However, the ion channel activity of CLICs need to be further confirmed under physiological conditions."

2. Fig 1: The authors state that IAA94 is an inhibitor for the CLIC proteins. This statement is too specific for the CLICs as IAA94 can be an inhibitor for other chloride ion channels. Also it would be important to study the other chloride ion channel blockers that were tested on the function of CLIC proteins, such as A9C and DIDS. As previous studies have shown that IAA94 and A9C were able to block the ion channel activity as well as the enzymatic function of CLIC1 protein, but not DIDS, this trio of inhibitors has discrimination value. Additional concerns center on the viability of cells that were treated with up to 150uM of IAA94 and also the viability of cells in which 3 CLIC proteins are knocked out or down. Some test for viability would be helpful.

Reply: Thanks very much for the comments and suggestions.

1) We agree with the reviewer that IAA94 might also have other targets, so we modified the statement as:"To assess the role of CLICs in NLRP3 inflammasome activation, we firstly examined whether indanyloxyacetic acid-94 (IAA94), which has shown inhibitory activity for CLICs³⁹, could inhibit NLRP3 inflammasome activation."

2) As suggested, we also tested whether A9C or DIDS could inhibit NLRP3 inflammasome activation or NLRP3 agonist-induced chloride efflux and found that A9C could inhibit both nigericin-induced IL-1 β production and chloride efflux. In contrast, DIDS could inhibit nigericin-induced IL-1 β production, but had no effects on chloride efflux, suggesting that DIDS might target downstream events of chloride efflux to block NLRP3 activation. These results were shown as supplementary Fig.6a-d in the revised manuscript.

3. As suggested, we also found that inhibition of the CLICs activity or expression had no effects on the viability of BMDMs in these condition (supplementary Fig.1d and supplementary Fig.3f in the revised manuscript).

3. Fig 4: Graphs A, B and even C show an intracellular chloride ion percentage more than 100% after 30 mins or 2.5 hours post-Nigericin, ATP and A:T treatment. Is this a rebounds effect or within the variation of the measurement? This is a bit confusing since the graphs do not show error bars and if we assume the error bars are small, then why do we see a small significant difference between the Mock and IAA94 20 mins post ATP treatment (graph E) but there is no significant difference between NLRP3 and its mutant after 2.5 hours treatment with poly A:T (graph C)?

Reply: Thanks for the comments. The errors bar were shown as mean \pm SEM, but they are small. We apologized for the miss labeling of significant differences in Fig. 4c (poly A:T treatment). These have been corrected in the revised manuscript. We did find significant difference for the rebounds effect of chloride between Nlrp3^{+/+} and Nlrp3^{-/-} cells post poly A:T treatment. The mechanisms for this need to be further investigated in future.

4. Fig 7: It would be useful to show the change in input under the same treatments as shown for the plasma membranes to judge overall changes for each protein. This set of experiments would also be enhanced if duplicated by CLIC protein immunofluorescence also, since it might detect where the translocated CLIC proteins travel from. Previously CLIC 1, 4 and 5 have been described in various cellular compartments including mitochondria so that the additional information gained from IIF could be very informative. A time course study is also essential to understand the dynamics of CLIC4 migrating to the plasma membrane and the rapid change in chloride seen over a matter of minutes.

Reply: Thanks very much for the suggestions and comments.

1) As suggested, we have showed the input under the same treatments for the plasma membranes (Fig.7a-c in the revised manuscript).

2) As suggested, we provided the immunofluorescence for the translocation of endogenous Clic4 in the revised manuscript (Fig.7d-f) in the revised manuscript. Because we didn't find antibodies against Clic3 or Clic1 working for immunofluorescence, we could not provide the data and hope the reviewer could understand.

3) We also provided the data about the time course of CLICs translocation (Fig.7a in the revised manuscript). The data showed that the translocation of CLICs during inflammasome activation is rapid.

5. Fig 8: Here again additional very informative information could be obtained if the authors showed that a CLIC protein was actually present in the NLRP3-NEK7 complex using their IP methods. Such a discovery would add a new dimension to our understanding of this protein family.

Reply: Thanks very much for the suggestion. As suggested, we performed Co-IP and found that CLICs were not present in NEK7-NLRP3 or NLRP3-ASC complex during inflammasome activation. The data were shown as supplementary Fig.5a, b in the revised manuscript.

6. Comments on the Discussion section:

For the sake of full disclosure, it would be important to reveal that, according to previous work, CLIC1 is a poorly anion selective ion channel and CLIC4 and CLIC5 are non selective ion channels with equal permeability to potassium and chloride. Therefore, it is important to include the possibility of other IAA94 sensitive chloride channels having involvement in this study.

To reiterate an alternative explanation for the results, an analysis of the time course of membrane translocation is very important. Depending on these data, the majority of the results in this paper may have been obtained from the non-membrane fractions of cells (e.g. IAA94 that was included in Fig1, was also able to block the enzymatic function of the soluble form of CLIC1 and presumably of the other CLIC proteins. Even the immunofluorescence imaging data provided does not show the CLICs in the membrane fractions of cells. Again it is critical to talk about the possibility of linking the enzymatic function of the CLIC proteins to NLRP3 activation and assembly process.

Reply: Thanks for the comments and suggestions. We have discussed the poor selectivity of CLICs, the possible involvement of other IAA94 sensitive chloride channels, the possible contribution of cytosolic CLICs (especially the enzymatic function of the CLICs) and also the possibility of CLICs as activators or modulators of chloride channels during NLRP3 activation in the Discussion section:

"Our study demonstrates NLRP3 agonists induce the translocation of CLICs to plasma membrane and robust chloride efflux *via* CLICs-dependent manner, suggesting that CLICs themselves might function as chloride channels to mediate NLRP3 inflammasome activation *via* promoting chloride efflux. Consistent with this, CLIC members have shown chloride channel activity when they exist as membrane form^{34, 35, 37, 44}. Although previous studies have shown that the channels formed by recombinant CLICs in artificial bilayers have poorly selectivity and are almost equally permeable by potassium and chloride^{45, 46}, suppression of the expression or activity of CLICs had no effect on potassium efflux or calcium influx (data not shown) during NLRP3 inflammasome activation, suggesting that CLICs might function as specific chloride channels under this condition. However, it should be noted that we cannot exclude the possibility that CLICs might function as an activator or modulator for a membrane chloride channel during inflammasome activation. Recently, the volume-regulated anion channel (VRAC) has been proposed to be involved in NLRP3 inflammasome activation because its chemical inhibitor can block NLRP3 inflammasome activation²³, suggesting CLICs might function as an activator for VRAC. So, if the role of VRAC in NLRP3 inflammasome activation could be confirmed by genetic evidences, the relationship between VRAC and CLICs needs to be clarified in future. In addition, although we have observed the translocation of CLICs to plasma membrane, CLICs-dependent chloride efflux and the inhibitory effects of IAA94 on IL-1 β production during NLRP3 inflammasome activation, the cytosolic CLICs might also contribute to NLRP3 inflammasome activation, because IAA94 can also inhibit the glutaredoxin-like enzymatic activity of soluble CLICs³¹. We then tested whether soluble CLICs were present in NLRP3 inflammasome complex during activation, but we didn't found the interactions between CLICs with the known NLRP3 inflammasome components, suggesting that CLICs or their glutaredoxin-like enzymatic activity cannot directly regulate the assembly of NLRP3 inflammasome. However, we still could not exclude the possible indirect contribution of cytosolic CLICs in NLRP3 inflammasome activation.

Although our results have shown that IAA94 can block NLRP3 inflammasome activation, suppression of the expression of CLICs cannot inhibit NLRP3 inflammasome activation completely. One possibility is that the family members are functionally redundant. Indeed, inhibition of all *Clic1*, *4* and *5* expression had much better inhibitory effects for inflammasome activation than inhibition of single *Clics* expression, possibly because the expression of other *Clics*

were compensatory increased. The redundant roles of CLIC family members have also been described in previous studies^{47, 48}. Another evidence for the functional redundancy of Clics was that deletion of two members of *Clic1*, *Clic4* or *Clic5* caused embryonic lethality, but deletion of one member did not, suggesting that CLICs are redundant in embryonic development. Another possibility is that other IAA94 sensitive chloride channels are also involved in NLRP3 inflammasome activation."

7. Finally, there are numerous typographical errors in the manuscript that will require careful editorial corrections.

Reply: We apologized for the errors. The manuscript has been examined carefully and the errors have been corrected.

Reviewer #3 (Remarks to the Author):

Tang et al. address the role of chloride intracellular channels (CLIC's) in Nlrp3 inflammasome activation and report that CLIC's play a crucial role downstream of K⁺ and ROS signaling but upstream to NEK7-Nlrp3 complex formation and caspase-1 activation. They also demonstrate that subsequent to mitochondrial ROS generation, CLIC's translocate to the plasma membrane promoting chloride efflux, which results in NEK7-Nlrp3-ASC association. The study is well performed and conclusion appropriately drawn. I do however have a few concerns:

1) LPS induced NFkB and IRF3 were shown to upregulate the expression levels of clic4 (abundantly expressed on macrophages) in major organs employing a positive feed forward loop (Ref 28) but the data presented in figure S2A contradicts or shows minimal effects on the expression levels. How do authors explain this and would it be possible for authors to present protein levels of clic1/4/5 under LPS-priming and NLRP3 inflammasome activation conditions.

Reply: Thanks for the comments and suggestion.

1) We did find that LPS could upregulate the expression of Clic in BMDMs (supplementary Fig.2a). LPS stimulated BMDMs had much higher level of Clic4 mRNA compared with control (about 7 fold, the scale of Y axis is Log10).

2) As suggested, we also provided the protein levels clic1/4/5 under LPS-priming and NLRP3 inflammasome activation conditions (supplementary Fig.2b, c).

2) Do soluble CLIC's present in the cytosol interact with NEK7-ASC-Nlrp3 complex?

Reply: Thanks very much for the suggestions. As suggested, we performed Co-IP and found that CLICs were not present in NEK7-NLRP3 or NLRP3-ASC complex during inflammasome activation. The data were shown as supplementary Fig.5a, b in the revised manuscript.

3) The specificity of IAA94 is a concern as it potentially blocks K⁺ and Ca²⁺ signaling in addition to CLICs. This should be discussed.

Reply: Thanks very much for the suggestion. We tested this and found that IAA94 had no effects on ATP-induced potassium (supplementary Fig.8a) or calcium efflux (see the figure below) during NLRP3 activation.

We also discussed this issue: "Although previous studies have shown that the channels formed by recombinant CLICs in artificial bilayers have poor selectivity and are almost equally

permeable by potassium and chloride^{45, 46}, suppression of the expression or activity of CLICs had no effect on potassium efflux or calcium influx (data not shown) during NLRP3 inflammasome activation, suggesting that CLICs might function as specific chloride channels under this condition."

4) Data from figures 2 and 3 suggest that CLIC4 & 5 play a dominant role. Have the authors looked at *Clic4*^{-/-} macrophages in which either *Clic1* or *Clic5* have been singly targeted with siRNA. Although CLIC1/4 double knockouts are embryonically lethal, is this also the case for CLIC4/5 or CLIC1/5? I realize the generation of new double knockout mice may be beyond the scope of the current study the authors should at least try to address this with an in vitro siRNA approach.

Reply: Thanks very much for the suggestion.

1) As suggested, we provided data showing that knockdown the expression of single *Clic1* or *Clic5* expression had better inhibitory activity for NLRP3 inflammasome in *Clic4*^{-/-} BMDMs (Fig.2c).

2) We have also been trying to get *Clic4/5* or *Clic1/5* KO mice, but these mice are also embryonically lethal. Actually, we have also provided showing that inhibition of the other two *Clics* expression in *Clic1* or *Clic5* KO BMDMs could suppress NLRP3 activation (Fig.2f, g).

5) Why didn't the authors use siRNA against CLIC1 in their in vivo model (figure3)? How do authors justify their claim of in vivo data with regard to compensatory effects of CLIC1 overexpression (figure S2E) in the absence of CLIC4?

Reply: Thanks very much for the comments. Actually, we also tried to inhibit the expression of *Clic1* in vivo by siRNA using nanoparticle, but the knockdown effects were not good. Based on the data, knockdown of *Clic5* in *Clic4* KO cells only partially suppressed MSU-induced peritoneal inflammation (Fig.3d, e), suggesting the compensatory effects of *Clic1* in the absence of *Clic4*.

6) In figure 5G the chloride-free buffers induced IL-1b independent of CLICs leading to the hypothesis that other chloride channels are involved. This should be substantiated by evaluating intracellular Cl⁻ levels to demonstrate a Cl⁻ efflux under these conditions. In addition, incubation of BMDMs (Fig S6C) in Cl⁻ free buffers did not induce any mitochondrial ROS production; do the authors propose that Cl⁻ efflux is not only required but also sufficient for NLRP3 inflammasome activation.

Reply: Thanks for the suggestion and comments.

1) As suggested, we provided the data showing that chloride-free buffers could induce chloride efflux (supplementary Fig.7c in the revised manuscript).

2) Based on the data from "chloride free buffers incubation", it seems chloride efflux is sufficient to activate NLRP3 inflammasome.

7) The authors should provide additional details in the methods section regarding the specifics for timing of addition of pharmacologic inhibitors and for the co-immunoprecipitation technique so these studies can be replicated by others.

Reply: Thanks for the suggestions. We have provided details for these protocols in the **Methods** section.

REVIEWERS' COMMENTS:

Reviewer #1 (Remarks to the Author):

I am satisfied with the rebuttal.

Reviewer #2 (Remarks to the Author):

no further comments

Reviewer #3 (Remarks to the Author):

The authors have addressed my previous concerns. I have only one comment: the authors might consider including the data showing IAA94 treatment doesn't affect Ca²⁺ signaling (reviewer 3 Point#3) in supplement information as this is particularly relevant for the current study as previously calcium was shown to regulate CLIC4 expression in mouse and human keratinocytes.

Reply: As suggested, we put the data about the role of IAA94 on calcium in supplementary Fig.9e.